# Towards Achieving Adversarial Robustness Beyond Perceptual Limits

Sravanti Addepalli [* 1]   Samyak Jain [* 2]   Gaurang Sriramanan [1]   Shivangi Khare [1]   R. Venkatesh Babu [1]

## Abstract

The vulnerability of Deep Neural Networks to Adversarial Attacks has fuelled research towards building robust models. While most existing Adversarial Training algorithms aim towards defending against imperceptible attacks, real-world adversaries are not limited by such constraints. In this work, we aim to achieve adversarial robustness at larger epsilon bounds. We first discuss the ideal goals of an adversarial defense algorithm beyond perceptual limits, and further highlight the shortcomings of naively extending existing training algorithms to higher perturbation bounds. In order to overcome these shortcomings, we propose a novel defense, Oracle-Aligned Adversarial Training (OA-AT), that attempts to align the predictions of the network with that of an Oracle during adversarial training. The proposed approach achieves state-of-the-art performance at large epsilon bounds ($\ell_\infty$ bound of $16/255$) while outperforming adversarial training algorithms such as AWP, TRADES and PGD-AT at standard perturbation bounds ($\ell_\infty$ bound of $8/255$) as well.

## 1. Introduction

Deep Neural Networks are known to be vulnerable to Adversarial Attacks, which are perturbations crafted with an intention to fool the network [25]. In a classification setting, adversarially perturbed images cause the network prediction to flip to unrelated classes, while causing no change in a human's prediction (Oracle label). The definition of adversarial attacks involves the presence of an Oracle, and this makes it challenging to formalize threat models for the training and verification of adversarial defenses. The widely accepted convention used in practice is the $\ell_p$ norm based threat model [5] with low-magnitude bounds to ensure im-

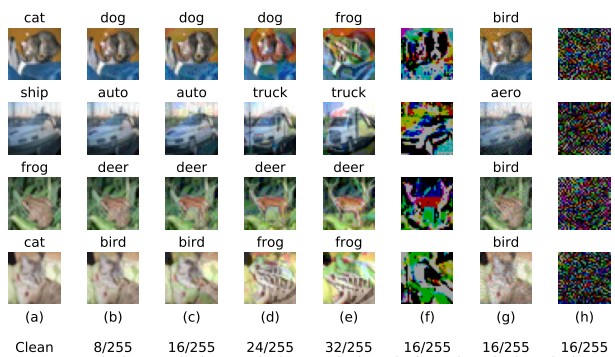

*Figure 1.* Adversarially attacked images (b-e, g) and perturbations (f, h) for various $\ell_\infty$ bounds. Attacks are generated from a PGD Adversarially Trained model (AT) [18, 19] or a Normally Trained model (NT). Original unperturbed image is shown in (a). Prediction of the attack source model is printed above each image.

perceptibility [11]. For example, attacks constrained within an $\ell_\infty$ norm of $8/255$ on the CIFAR-10 dataset are imperceptible to the human eye as shown in Fig.1(b), ensuring that the Oracle label is unchanged.

While low-magnitude $\ell_p$ norm based threat models form a crucial subset of the widely accepted definition of adversarial attacks [10], they are not sufficient, as there exist valid attacks at higher $\varepsilon$-bounds as well, as shown in Fig.1(g). However, the challenge at large perturbation bounds is the existence of attacks that can flip Oracle labels as well [28], as shown in Fig.1(c-e). This makes it difficult to naively scale existing Adversarial Training algorithms to large $\varepsilon$ bounds. In this work, we aim to improve robustness at larger epsilon bounds, such as an $\ell_\infty$ norm bound of $16/255$. We define this as a moderate-magnitude bound, and discuss the ideal goals for achieving robustness under this threat model in Sec.3. We further propose a novel defense Oracle-Aligned Adversarial Training (OA-AT), which attempts to align the predictions of the network with that of an Oracle, rather than enforcing all samples within the constraint set to have the same label as the unperturbed image.

Our contributions have been summarized below:

- We define the ideal goals for a moderate-$\varepsilon$ threat model ($\ell_\infty$ radius of $16/255$) and construct our goals as a feasible subset of the same.
- We propose methods for generating Oracle-Aligned adversaries, which can be used for adversarial training.

*Equal contribution  [1]Video Analytics Lab, Department of Computational and Data Sciences, Indian Institute of Science, Bangalore, India [2]Department of Computer Science and Engineering, Indian Institute of Technology (BHU), Varanasi, India. Correspondence to: Sravanti Addepalli <sravantia@iisc.ac.in>.

*Accepted by the ICML 2021 workshop on A Blessing in Disguise: The Prospects and Perils of Adversarial Machine Learning.* Copyright 2021 by the author(s).

- We propose Oracle-Aligned Adversarial Training (OA-AT) to improve robustness within the defined moderate-$\varepsilon$ threat model.

- We demonstrate superior performance when compared to state-of-the-art methods such as AWP [30], TRADES [31] and PGD-AT [18, 19] at $\varepsilon = 16/255$ while also performing better at $\varepsilon = 8/255$.

- We achieve improvements over the baselines even at larger model capacities such as ResNet-34 and WideResNet-34-10.

## 2. Related Works

**Robustness against imperceptible attacks:** Adversarial Training has emerged as the most successful defense strategy against $\ell_p$ norm bound imperceptible attacks. PGD Adversarial Training (PGD-AT) [18] constructs multi-step adversarial attacks by maximizing Cross-Entropy loss within the considered threat model and subsequently minimizes the same for training. This was followed by several adversarial training methods [31, 1, 20, 30, 23, 19] that improved accuracy against such imperceptible threat models further. Zhang et al. [31] proposed the TRADES defense, which maximizes the Kullback-Leibler (KL) divergence between the softmax outputs of adversarial and clean samples for attack generation, and minimizes the same in addition to the Cross-Entropy loss on clean samples for training.

**Improving Robustness of base defenses:** Wu et al. [30] proposed an additional step of *Adversarial Weight Perturbation* (AWP) to perturb the weights of the model in order to maximize the training loss, and further train the perturbed model to minimize the same. This generates a flatter loss surface [24], thereby improving robust generalization. While this can be integrated with any defense, AWP-TRADES is the state-of-the-art adversarial defense today. On similar lines, the use of stochastic weight averaging of model weights [14] is also seen to improve the flatness of loss surface, resulting in a boost in robustness [12, 6]. Recent works [19, 20, 12] attempt to find the best training techniques such as early stopping, use of optimal weight decay and weight averaging to achieve enhanced robust performance on base defenses such as PGD-AT [18] and TRADES [31].

**Robustness against large perturbation attacks:** Shaeiri et al. [21] demonstrate that the standard formulation of adversarial training is not well-suited for achieving robustness at large perturbations, as the loss saturates very early. The authors propose Extended Adversarial Training (ExAT), where a model trained on low-magnitude perturbations ($\varepsilon = 8/255$) is fine-tuned with large magnitude perturbations ($\varepsilon = 16/255$) for merely 5 training epochs, to achieve improved robustness at large perturbations. The authors also discuss the use of a varying epsilon schedule to improve training convergence. Friendly Adversarial Training

(FAT) [1] performs early-stopping of an adversarial attack by thresholding the number of times the model misclassifies the image during attack generation. The threshold is increased over training epochs to increase the strength of the attack over training. On similar lines, Sitawarin et al. [22] propose Adversarial Training with Early Stopping (ATES), which performs early stopping of a PGD attack based on the margin of the perturbed image being greater than a threshold that is increased over epochs. We improve upon these methods significantly using our proposed approach (Sec.4).

## 3. Preliminaries and Threat Model

### 3.1. Notation

We consider an $N$-class image classification problem with access to a labelled training dataset $\mathcal{D}$. The input images are denoted by $x \in \mathcal{X}$ and their corresponding labels are denoted as $y \in \{1, ..., N\}$. The function represented by the Deep Neural Network is denoted by $f_\theta$ where $\theta \in \Theta$ denotes the set of trained network parameters. The $N$-dimensional softmax output of the input image $x$ is denoted as $f_\theta(x)$. Adversarial examples are defined to be images that are crafted specifically to fool a model into making an incorrect prediction [10]. An adversarial image corresponding to a clean image $x$ would be denoted as $\widetilde{x}$. The set of all images within an $\ell_p$ norm ball of radius $\varepsilon$, $\mathcal{S}(x)$ is defined as, $\mathcal{S}(x) = \{\hat{x} : ||\hat{x} - x||_p < \varepsilon\}$. The set of all $\ell_p$ norm bound adversarial examples, $\mathcal{A}(x)$ is defined as, $\mathcal{A}(x) = \{\widetilde{x} : f_\theta(\widetilde{x}) \neq y, \widetilde{x} \in \mathcal{S}(x)\}$. In this work, we specifically consider robustness to $\ell_\infty$ norm bound adversarial examples. We define the Oracle prediction of a sample $x$ as the label that a human is likely to assign to the image, and denote it as $O(x)$. For a clean image, $O(x)$ would correspond to the true label $y$, while for a perturbed image it could differ from the original label.

### 3.2. Nomenclature of Adversarial Attacks

Tramèr et al. [28] discuss the existence of two types of adversarial examples: Sensitivity-based examples, where the model prediction changes, but the Oracle prediction remains the same as the unperturbed image, and Invariance-based examples, where the Oracle prediction changes, while the model prediction remains unchanged. Models trained using standard empirical risk minimization are susceptible to sensitivity-based adversarial examples, while models which are overly robust to large perturbation bounds could be susceptible to invariance-based examples. Since these definitions are dependent on the model being considered, we define a different nomenclature which only depends on the input image and the threat model considered, as below:

- Oracle-Invariant set $OI(x)$, is defined as the set of all images within the bound $\mathcal{S}(x)$, which preserve the Oracle label. The Oracle is invariant to such perturbations: $OI(x) := \{\hat{x} : O(\hat{x}) = O(x), \hat{x} \in \mathcal{S}(x)\}$

- Oracle-Sensitive set $OS(x)$, is defined as the set of all images within the bound $\mathcal{S}(x)$, which flip the Oracle label. The Oracle is sensitive to such perturbations: $OS(x) := \{\hat{x} : O(\hat{x}) \neq O(x), \hat{x} \in \mathcal{S}(x)\}$

### 3.3. Objectives of the Proposed Defense

Defenses based on the conventional $\ell_p$ norm based threat model defined in Sec.3.1 attempt to train models which are invariant to all samples within $\mathcal{S}(x)$. This is an ideal requirement for low $\varepsilon$-bound perturbations, where the added noise is imperceptible, and hence all samples within the threat model are Oracle-Invariant. An example of a low $\varepsilon$ threat model is the constraint set defined by $\varepsilon = 8/255$ for the CIFAR-10 dataset, which produces adversarial examples that are perceptually similar to the corresponding clean images, as shown in Fig.1(b).

As we move to larger $\varepsilon$ bounds, Oracle-labels begin to change, as shown in Fig.1(c, d, e). For a very high perturbation bound such as $32/255$, the changes produced by an attack are clearly perceptible and cause a change in the Oracle label in many cases. Hence, robustness at such large bounds may not be of much practical relevance. The focus of this work is to achieve robustness within a moderate-magnitude $\ell_p$ norm bound threat model, where some perturbations look partially modified (Fig.1(c)), while others look unchanged (Fig.1(g)), as is the case with $\varepsilon = 16/255$ for CIFAR-10. The existence of attacks that do not significantly change the perception of the image necessitates the requirement of robustness within such bounds, while the existence of partially Oracle-Sensitive samples makes it difficult to use standard adversarial training methods on the same. The ideal goals for training defenses under this moderate-magnitude threat model are described below:

- Robustness against samples which belong to $OI(x)$
- Sensitivity towards samples which belong to $OS(x)$, with model's prediction matching the Oracle label
- No specification on Out-of-Distribution (OOD) images

We incorporate these goals in the training objective of our proposed defense, which is discussed in Sec.4. Given the practical difficulty in assigning Oracle labels, we consider the following criteria for our defense evaluations:

- Robustness-Accuracy trade-off, measured using accuracy on clean samples and robustness against valid attacks within the threat model (discussed below)
- Robustness against all attacks within $\varepsilon = 8/255$, measured using strong white-box attacks [8, 23]
- Robustness to Oracle-Invariant samples within $\varepsilon = 16/255$, measured using gradient-free attacks [2]

We do not explicitly define goals for white-box attacks within the moderate $\varepsilon$ bound of $16/255$ since the existence of Oracle-Sensitive samples within this bound is image specific. We note from Fig.1(c) and Fig.A2(b) that most adversarial examples look partially modified at $\varepsilon = 16/255$.

## 4. Proposed Method

In order to achieve the goals discussed in Sec.3.3, we require to generate Oracle-Sensitive and Oracle-Invariant samples and impose specific training losses on each of them individually. Since the labeling of adversarial samples as Oracle-Invariant or Oracle-Sensitive is expensive and cannot be done while training networks, we propose to use attacks which ensure a given type of perturbation (OI or OS) by construction, and hence do not require explicit annotation.

**Generation of Oracle-Sensitive examples:** Robust models are known to have perceptually aligned gradients [29]. Adversarial examples generated using a robust model tend to start looking like the target (other) class images at large perturbation bounds, as seen in Fig.1(c, d, e). We therefore use large $\varepsilon$ white-box adversarial examples generated from the model being trained as Oracle-Sensitive samples, and the model prediction as a proxy to the Oracle prediction.

**Generation of Oracle-Invariant examples:** While the strongest Oracle-Invariant examples are generated using the gradient-free Square attack [2], it uses 5000 queries, which is computationally expensive for use in adversarial training. Reducing the number of queries weakens the attack significantly. The most efficient attack that is widely used for adversarial training is the PGD 10-step attack. However, it cannot be used for the generation of Oracle-Invariant samples as gradient-based attacks generated from adversarially trained models produce Oracle-Sensitive samples. We propose to use the Learned Perceptual Image Patch Similarity (LPIPS) measure for the generation of Oracle-Invariant attacks, as it is known to match well with perceptual similarity [32, 17]. As shown in Fig.A1, while the standard AlexNet model used in prior work [17] fails to distinguish between Oracle-Invariant and Oracle-Sensitive samples, an adversarially trained model is able to distinguish between the two effectively. We therefore propose to minimize the LPIPS distance between natural and perturbed images, in addition to the maximization of Cross-Entropy loss for attack generation: $\mathcal{L}_{CE}(x, y) - \lambda \cdot \text{LPIPS}(x, \hat{x})$. We choose $\lambda$ as the minimum value that transforms the attack from Oracle-Sensitive to Oracle-Invariant (OI), to generate strong OI attacks (Fig.A2). This is further fine-tuned during training to achieve the optimal robustness-accuracy trade-off.

**Oracle-Aligned Adversarial Training (OA-AT):** The training algorithm for the proposed defense, Oracle-Aligned Adversarial Training (OA-AT) is presented in Algorithm-A1. We use the Trades-AWP formulation [31, 30] as the base implementation, with Cross-Entropy loss instead of KL-divergence loss for attack generation, as it results in stronger attacks [12]. Different from Wu et al. [30], we maximize loss on $x_i + 2 \cdot \widetilde{\delta}_i$ (where $\widetilde{\delta}_i$ is the attack) in the additional weight perturbation step, as it results in improved robust generalization. We use cosine learning rate schedule.

*Table 1.* **CIFAR-10, CIFAR-100**: Performance (%) of the proposed defense OA-AT compared to baselines, against attacks with different $\varepsilon$ bounds. Sorted by AutoAttack accuracy [8] (AA 8/255)

| Method | Clean | GAMA (8/255) | AA (8/255) | GAMA (12/255) | Square (12/255) | GAMA (16/255) | Square (16/255) |
|---|---|---|---|---|---|---|---|
| CIFAR-10 (ResNet-18) | | | | | | | |
| FAT [1] (8) | **84.36** | 48.41 | 48.14 | 29.39 | 39.48 | 15.18 | 25.07 |
| PGD-AT [18] (10) | 79.38 | 49.28 | 48.68 | 32.40 | 41.46 | 18.18 | 28.29 |
| AWP [30] (10) | 80.32 | 49.06 | 48.89 | 32.88 | 40.27 | 19.17 | 27.56 |
| ATES [22] (10) | 80.95 | 49.57 | 49.12 | 32.44 | 42.21 | 18.36 | 29.07 |
| TRADES [31] (8) | 80.53 | 49.63 | 49.42 | 33.32 | 40.94 | 19.27 | 27.82 |
| ExAT-PGD [21] (11) | 80.68 | 50.06 | 49.52 | 32.47 | 41.10 | 17.81 | 27.23 |
| ExAT + AWP (10) | 80.18 | 49.87 | 49.69 | 33.51 | 41.04 | 20.04 | 28.40 |
| AWP [30] (8) | 80.47 | 50.06 | 49.87 | 33.47 | 41.05 | 19.66 | 28.51 |
| **OA-AT (Ours)** | 80.24 | **51.40** | **50.88** | **36.01** | **43.20** | **22.73** | **31.16** |
| Gain w.r.t. AWP | −0.23 | +1.34 | +1.01 | +2.54 | +2.15 | +3.07 | +2.65 |
| CIFAR-100 (ResNet-18) | | | | | | | |
| AWP [30] | 59.88 | 25.81 | 25.52 | 14.80 | 20.24 | 8.72 | 12.80 |
| **OA-AT (Ours)** | **60.27** | **26.41** | **26.00** | **16.28** | **21.47** | **10.47** | **14.60** |
| Gain w.r.t. AWP | +0.39 | +0.60 | +0.48 | +1.48 | +1.23 | +1.75 | +1.80 |

*Table 2.* **CIFAR-10**: Performance (%) of the proposed defense OA-AT (Ours) compared to the strongest baseline, AWP-TRADES (AWP) [30] against various attacks with different $\varepsilon$ bounds

| Method | Model | Clean | AA (8/255) | Square (12/255) | AA (12/255) | Square (16/255) | AA (16/255) |
|---|---|---|---|---|---|---|---|
| AWP | RN-18 | 80.47 | 49.87 | 41.05 | 33.19 | 28.51 | 19.23 |
| **Ours** | RN-18 | 80.24 | 50.88 | 43.20 | 35.39 | 31.16 | 22.00 |
| AWP | RN-34 | 83.89 | 52.44 | 42.84 | 34.61 | 29.22 | 19.69 |
| **Ours** | RN-34 | 84.07 | 53.22 | 45.03 | 36.31 | 32.47 | 22.00 |
| AWP | WRN-34 | 85.19 | 55.69 | 46.48 | 38.05 | 32.68 | 23.46 |
| **Ours** | WRN-34 | 85.54 | 55.67 | 48.15 | 38.13 | 35.20 | 22.92 |
| AWP + WA | WRN-34 | 85.10 | 55.87 | 46.52 | 37.97 | 32.50 | 23.27 |
| **Ours + WA** | WRN-34 | **85.67** | **55.93** | **48.79** | **39.06** | **35.76** | **24.05** |

We start with an initial $\varepsilon$ value of $4/255$ upto one-fourth the training epochs, and ramp up this value linearly to a value of $16/255$ at the last epoch. We use 5 attack steps when $\varepsilon = 4/255$ and 10 attack steps later. We perform standard adversarial training upto $\varepsilon = 12/255$ as the attacks in this range are imperceptible. Beyond this, we start incorporating separate training losses for Oracle-Invariant and Oracle-Sensitive samples in alternate training iterations. Oracle-Sensitive samples are generated by maximizing Cross-Entropy loss in a PGD attack formulation. Rather than enforcing the predictions of such attacks to be similar to the original image, we allow the network to be partially sensitive to such attacks by training them to be similar to a convex combination of predictions on the clean image and perturbed samples at larger ($1.5 \cdot \varepsilon_{max}$) bounds as shown:

$$L_{adv} = KL\big(f_\theta(x_i + \widetilde{\delta_i}) \,\|\, \alpha \, f_\theta(x_i) + (1-\alpha) \, f_\theta(x_i + \widehat{\delta_i})\big)$$

Here $\widetilde{\delta_i}$ is the perturbation at the varying epsilon value $\widetilde{\varepsilon}$, and $\widehat{\delta_i}$ is the perturbation at $24/255$. This results in better robustness-accuracy trade-off as shown in Table-A1. In the other alternate iteration, we use the LPIPS metric to generate strong and efficient Oracle-Invariant attacks during training. We perform exponential weight-averaging of the network being trained and use this for computing the LPIPS metric for improved and stable results (Table-A1). We increase $\alpha$ and $\lambda$ over training, as the nature of attacks changes with varying $\varepsilon$. The use of both Oracle-Invariant (OI) and Oracle-Sensitive (OS) samples ensures robustness to OI samples while allowing sensitivity to partially OS samples.

## 5. Experiments and Results

We compare performance of the proposed approach with the existing defenses discussed in Sec.2 on the CIFAR-10 [16] dataset in Table-1. We train all models on ResNet-18 architecture for 110 epochs. For each baseline, we find the best set of hyperparameters to achieve clean accuracy of around $80\%$ to ensure a fair comparison across all meth-

ods. We also perform baseline training across various $\varepsilon$ values and report the best baselines in Table-1. We observe that baseline defenses do not perform well when trained using large $\varepsilon$ bounds such as $16/255$ (Table-A2). We report adversarial robustness against the strongest known attacks, AutoAttack (AA) [8] and GAMA PGD-100 (GAMA) [23] for $\varepsilon = 8/255$ in order to obtain the worst-case robust accuracy. For larger bounds such as $12/255$ and $16/255$, we primarily aim for robustness against the Square attack [2], as it is the strongest known Oracle-Invariant attack. We compare the proposed approach against the strongest baseline AWP-TRADES [30] on CIFAR-100 in Table-1 (ref. Table-A3 for detailed results), and on CIFAR-10 with larger capacity models in Table-2. We observe significant gains with the use of AutoAugment [9, 24] on CIFAR-100, and additionally with Model Weight Averaging (WA) [14, 12, 6] at larger model capacities. To ensure a fair comparison, we consider these for the AWP baseline as well.

**Results:** The proposed defense achieves consistent gains across all metrics considered in Sec.3.3 (AutoAttack [8] at $\varepsilon = 8/255$ and Square attack [2] at larger $\varepsilon$ bounds). Although we train the model for achieving robustness at larger $\varepsilon$ bounds, we achieve an improvement in the robustness at $\varepsilon = 8/255$ as well, which is not observed in any of the existing methods (Table-A2). We evaluate the proposed defense against diverse attacks (Table-A4) and sanity checks (Sec.A5) to ensure the absence of gradient masking.

## 6. Conclusions

We explore the idea of robustness beyond perceptual limits in an $\ell_p$ norm based threat model. We first discuss the ideal goals of an adversarial defense at larger perturbation bounds, and further propose a novel defense, Oracle-Aligned Adversarial Training (OA-AT) that aims to align model predictions with that of an Oracle during training. The key aspects of the defense include the use of LPIPS metric for generating Oracle-Invariant attacks during training, and the use of a convex combination of clean and adversarial image predictions as targets for Oracle-Sensitive samples. We achieve significant gains in robustness at low and moderate perturbation bounds, and a better robustness-accuracy trade-off.

# 7. Acknowledgements

This work was supported by Uchhatar Avishkar Yojana (UAY) project (IISC_10), MHRD, Govt. of India. Sravanti Addepalli is supported by a Google PhD Fellowship in Machine Learning. We are thankful for the support.

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

# Appendix

## A1. Oracle-Invariant Attacks

**Square Attack:** The strongest Oracle-Invariant examples are generated using the Square attack [2]. These images are Oracle-Invariant since this is a query-based attack and does not use gradients from any model for attack generation. However this attack uses 5000 queries, and is a computationally expensive attack. Hence this attack cannot be used for adversarial training, although it is one of the best attacks for evaluations. We note that reducing the number of queries makes it computationally efficient, however it also reduces the effectiveness of the attack significantly.

**PGD based Attacks:** While the most efficient attack that is widely used for adversarial training is the PGD 10-step attack, it cannot be used for the generation of Oracle-Invariant samples as adversarially trained models have perceptually aligned gradients, and tend to produce Oracle-Sensitive samples. Therefore, we explore some variants of the PGD attack to make the generated perturbations Oracle-Invariant. We denote the Cross-Entropy loss on a data sample $x$ with ground truth label $y$ using $\mathcal{L}_{CE}(x, y)$. We explore the addition of regularizers to the Cross-Entropy loss weighted by a factor of $\lambda_X$ in each case. The value of $\lambda_X$ is chosen as the minimum value which transforms the PGD attacks from Oracle-Sensitive to Oracle-Invariant. This results in the strongest possible Oracle-Invariant attacks.

**Discriminator based PGD Attack:** We train a discriminator to distinguish between Oracle-Invariant and Oracle-Sensitive adversarial examples, and further maximize the below loss for the generation of Oracle-Invariant attacks:

$$\mathcal{L}_{CE}(x, y) - \lambda_{Disc} \cdot \mathcal{L}_{BCE}(\hat{x}, \text{OI}) \qquad \text{(A1)}$$

Here $\mathcal{L}_{BCE}(\hat{x}, \text{OI})$ is the Binary Cross-Entropy loss of the adversarial example $\hat{x}$ w.r.t. the label corresponding to an Oracle-Invariant (OI) attack. We train the discriminator to distinguish between two input distributions; the first corresponding to images concatenated channel-wise with their respective Oracle-Sensitive perturbations, and a second distribution where perturbations are shuffled across images in the batch. This ensures that the discriminator relies on the spatial correlation between the image and its corresponding perturbation for the classification task, rather than the properties of the perturbation itself. The attack in Eq.A1 therefore attempts to break the most salient property of Oracle-Sensitive attacks, which is the spatial correlation between an image and its perturbation.

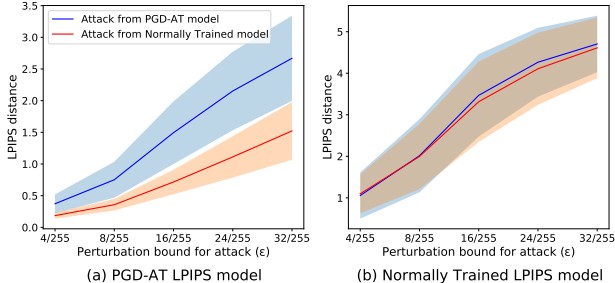

(a) PGD-AT LPIPS model     (b) Normally Trained LPIPS model

*Figure A1.* LPIPS distance between clean and adversarially perturbed images. Attacks generated from PGD-AT [18, 19] model (Oracle-Sensitive) and Normally Trained model (Oracle-Invariant) are considered. (a) PGD-AT ResNet-18 model is used for computation of LPIPS distance (b) Normally Trained AlexNet model is used for computation of LPIPS distance. PGD-AT model based LPIPS distance is useful to distinguish between Oracle-Sensitive and Oracle-Invariant attacks.

**LPIPS based PGD Attack:** We propose to use the Learned Perceptual Image Patch Similarity (LPIPS) measure for the generation of Oracle-Sensitive attacks, as it is known to match well with perceptual similarity [32, 17]. As shown in Fig.A1, while the standard AlexNet model that is used in prior work [17] fails to distinguish between Oracle-Invariant and Oracle-Sensitive samples, an adversarially trained model is able to distinguish between the two types of attacks effectively. In this plot, we consider attacks generated from a PGD-AT [18, 19] model (Fig.1(c-e)) as Oracle-Sensitive attacks, and attacks generated from a Normally Trained model (Fig.1(h)) as Oracle-Invariant attacks. We therefore propose to minimize the LPIPS distance between the natural and perturbed images, in addition to the maximization of Cross-Entropy loss for attack generation as shown below:

$$\mathcal{L}_{CE}(x, y) - \lambda_{\text{LPIPS}} \cdot \text{LPIPS}(x, \hat{x}) \qquad \text{(A2)}$$

We choose $\lambda_{\text{LPIPS}}$ as the minimum value that transforms the PGD attack from Oracle-Sensitive to Oracle-Invariant (OI), to generate strong OI attacks. This is further fine-tuned during training to achieve the optimal robustness-accuracy trade-off. As shown in Fig.A2, setting $\lambda_{\text{LPIPS}}$ to 1 changes adversarial examples from Oracle-Sensitive to Oracle-Invariant, as they look similar to the corresponding original images shown in Fig.A2(a). This can be observed more distinctly at perturbation bounds of $24/255$ and $32/255$. The perturbations in Fig.A2(c) are smooth, while those in (e) and (g) are not. This shows that the addition of

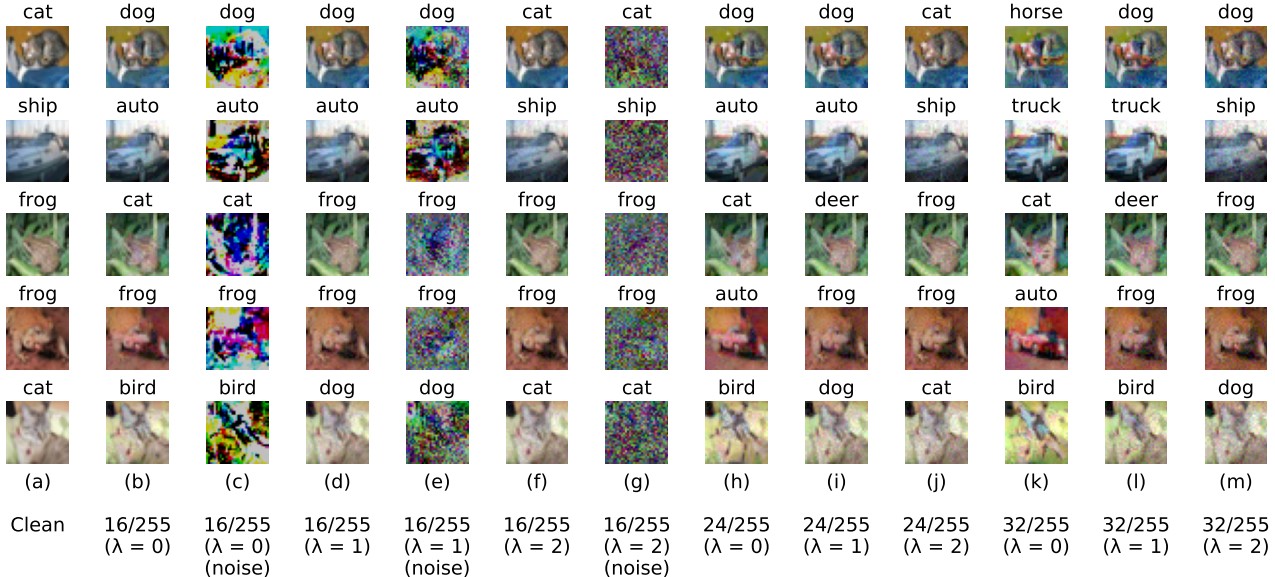

*Figure A2.* Oracle-Invariant adversarial examples generated using the LPIPS based PGD attack in Eq.A2 across various perturbation bounds. White-box attacks and predictions on the model trained using the proposed OA-AT defense on the CIFAR-10 dataset with ResNet-18 architecture are shown: (a) Original Unperturbed image, (b, h, k) Adversarial examples generated using the standard PGD 10-step attack, (d, f, i, j, l, m) LPIPS based PGD attack generated within perturbation bounds of $16/255$ (d, f), $24/255$ (i, j) and $32/255$ (l, m) by setting the value of $\lambda_{\text{LPIPS}}$ to 1 and 2, (c, e, g) Perturbations corresponding to (b), (d) and (f) respectively.

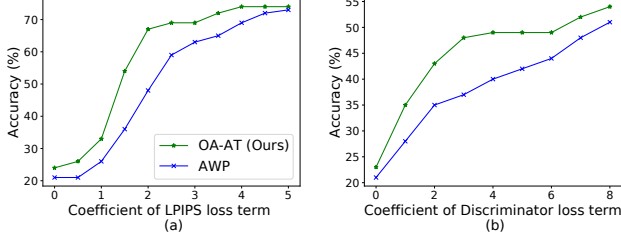

*Figure A3.* Comparison of the proposed model with the best baseline, AWP [30] trained on CIFAR-10 with ResNet-18 architecture, against attacks of varying strength and Oracle sensitivity constrained within perturbation bound of $\varepsilon = 16/255$. (a) LPIPS based regularizer, and (b) Discriminator based regularizer are used for generating Oracle-Invariant attacks respectively. As the coefficient of the regularizer increases, the attack transforms from Oracle-Sensitive to Oracle-Invariant. The proposed method (OA-AT) achieves improved accuracy compared to AWP.

the LPIPS term helps in making the perturbations Oracle-Invariant. Very large coefficients of the LPIPS term make the attack weak as can be seen in Fig.A2(f, j, m) where the model prediction is same as the true label. We therefore set the value of $\lambda_{\text{LPIPS}}$ to 1 to obtain strong Oracle-Invariant attacks.

As shown in Table-A1, while we obtain the best results using the LPIPS based PGD attack for training (E1), the use of discriminator based PGD attack (E6) also results in a better robustness-accuracy trade-off when compared to E2, where there is no explicit regularizer to ensure the generation of

Oracle-Invariant attacks.

**Evaluation of the proposed defense against Oracle-Invariant Attacks:** We compare the performance of the proposed defense OA-AT with the strongest baseline AWP [30] against the two proposed Oracle-Invariant attacks, LPIPS based attack and Discriminator based attack in Fig.A3 (a) and (b) respectively. We vary the coefficient of the regularizers used in the generation of attacks, $\lambda_{Disc}$ (Eq.A1) and $\lambda_{\text{LPIPS}}$ (Eq.A2) in each of the plots. As we increase the coefficient, the attack transforms from Oracle-Sensitive to Oracle-Invariant. The proposed method (OA-AT) achieves improved accuracy when compared to the AWP [30] baseline.

## A2. Details on the Datasets used

We evaluate the proposed approach on the CIFAR-10 and CIFAR-100 [16] datasets. The two datasets consist of RGB images of spatial dimension $32{\times}32$, and contain 10 and 100 distinct classes respectively. CIFAR-10 is the most widely used benchmark dataset to perform a comparative analysis across different adversarial defense and attack methods. CIFAR-100 is a challenging dataset to achieve adversarial robustness given the large number of diverse classes that are interrelated. Each of these datasets consists of 50,000 training images and 10,000 test images. We split the original training set to create a validation set of 1,000 images in CIFAR-10 and 2,500 images in CIFAR-100. We ensure that

---

**Algorithm A1** Oracle-Aligned Adversarial Training

1: **Input:** Deep Neural Network $f_\theta$ with parameters $\theta$, Training Data $\{x_i, y_i\}_{i=1}^{M}$, Epochs $T$, Learning Rate $\eta$, Perturbation budget $\varepsilon_{max}$, Adversarial Perturbation function $A(x, y, \ell, \varepsilon)$ which maximises loss $\ell$
2: **for** epoch $= 1$ **to** $T$ **do**
3:     $\widetilde{\varepsilon} = \max\{\varepsilon_{max}/4, \varepsilon_{max} \cdot \text{epoch}/T\}$
4:     **for** $i = 1$ **to** $M$ **do**
5:        $\delta_i \sim U(-\min(\widetilde{\varepsilon}, \varepsilon_{max}/4), \min(\widetilde{\varepsilon}, \varepsilon_{max}/4))$
6:        **if** $\widetilde{\varepsilon} < 3/4 \cdot \varepsilon_{max}$ **then**
7:          $\ell = \ell_{CE}(f_\theta(x_i + \delta_i), y_i)$
8:          $\widetilde{\delta}_i = A(x_i, y_i, \ell, \widetilde{\varepsilon})$
9:          $L_{adv} = \text{KL}\left(f_\theta(x_i + \widetilde{\delta}_i) || f_\theta(x_i)\right)$
10:       **else if** $i \% 2 = 0$ **then**
11:         $\ell = \ell_{CE}(f_\theta(x_i + \delta_i), y_i)$
12:         $\widehat{\delta}_i = A(x_i, y_i, \ell, 1.5 \cdot \varepsilon_{max})$
13:         $\widetilde{\delta}_i = \Pi_\infty(\widehat{\delta}_i, \widetilde{\varepsilon})$
14:         $L_{adv} = \text{KL}\left(f_\theta(x_i + \widetilde{\delta}_i) \quad || \right.$
                     $\left. \alpha \cdot f_\theta(x_i) + (1 - \alpha) \cdot f_\theta(x_i + \widehat{\delta}_i)\right)$
15:       **else**
16:         $\delta_i \sim U(-\widetilde{\varepsilon}, \widetilde{\varepsilon})$
17:         $\ell = \ell_{CE}(f_\theta(x_i + \delta_i), y_i) - \text{LPIPS}(x_i, x_i + \delta_i)$
18:         $\widetilde{\delta}_i = A(x_i, y_i, \ell, \widetilde{\varepsilon})$
19:         $L_{adv} = \text{KL}\left(f_\theta(x_i + \widetilde{\delta}_i) || f_\theta(x_i)\right)$
20:       **end if**
21:       $L = \ell_{CE}(f_\theta(x_i), y_i) + L_{adv}$
22:       $\theta = \theta - \eta \cdot \nabla_\theta L$
23:     **end for**
24: **end for**

---

the validation split is balanced equally across all classes, and use the remaining images for training. To ensure a fair comparison, we use the same split for training the proposed defense as well as other baseline approaches. For both datasets, we consider the $\ell_\infty$ threat model of radius $8/255$ to be representative of imperceptible perturbations, that is, the Oracle label does not change within this set. Further, we consider the $\ell_\infty$ threat model of radius $16/255$ to investigate robustness within moderate magnitude perturbation bounds.

## A3. Details on Training

The algorithm for the proposed method as explained in Sec.4 is presented in Algorithm-A1. We use a varying $\varepsilon$ schedule and start training on perturbations of magnitude $\varepsilon = 4/255$. This results in marginally better performance when compared to ramping up the value of $\varepsilon$ from 0 (E8 of Table-A1). For CIFAR-10 training on ResNet-18, we set the weight of the adversarial loss $L_{adv}$ in L21 of Alg.A1 ($\beta$ parameter of TRADES [31]) to 1.5 for the first three-quarters of training, and then linearly increase it from 1.5 to 3 in the moderate perturbation regime, where $\varepsilon$ is linearly increased from

$12/255$ to $16/255$. In this moderate perturbation regime, we also linearly increase the coefficient of the LPIPS distance (Alg.A1, L17) from 0 to 1, and linearly decrease the $\alpha$ parameter used in the convex combination of softmax prediction (Alg.A1, L14) from 1 to 0.8. This results in a smooth transition from adversarial training on imperceptible attacks to attacks with larger perturbation bounds. We set the weight decay to 5e-4.

For all our experiments, we use the cosine learning rate schedule with 0.2 as the maximum learning rate. We use SGD optimizer with momentum of 0.9, and train for 110 epochs. We compute the LPIPS distance using an exponential weight averaged model with $\tau = 0.995$. We note from Table-A1 that the use of weight-averaged model results in better performance when compared to using the model being trained for the same (E5). This also leads to more stable results across reruns.

We utilise AutoAugment [9] for training on CIFAR-100, and for CIFAR-10 training on large model capacities. We apply AutoAugment with a probability of 0.5 for CIFAR-100, and for the CIFAR-10 model trained on ResNet-34. Since the extent of overfitting is higher for large model capacities, we use AutoAugment with $p = 1$ on WideResNet-34-10. While the use of AutoAugment helps in overcoming overfitting, it could also negatively impact robust accuracy due to the drift between the training and test distributions. We observe a drop in robust accuracy on the CIFAR-10 dataset with the use of AutoAugment (E11, E12 in Table-A1), while there is a boost in the clean accuracy. On similar lines, we observe a drop in robust accuracy on the CIFAR-100 dataset as well, when we increase the probability of applying AutoAugment from 0.5 (E11 in Table-A1) to 1 (E12 in Table-A1).

To investigate the stability of the proposed approach, we train a ResNet-18 network multiple times by using different random initialization of network parameters. We observe that the proposed approach is indeed stable, with standard deviation of 0.167, 0.115, 0.180 and 0.143 for clean accuracy, GAMA PGD-100 accuracies with $\varepsilon = 8/255$ and $16/255$, and accuracy against the Square attack with $\varepsilon = 16/255$ respectively over three independent training runs on CIFAR-10. We also observe that the last epoch is consistently the best performing model for the ResNet-18 architecture. Nonetheless, we still utilise early stopping on the validation set using PGD 7-step accuracy for all the baselines to enable a fair comparison overall.

### A3.1. Ablation Study

In order to study the impact of different components of the proposed defense, we present a detailed ablative study using ResNet-18 models in Table-A1. We present results on the CIFAR-10 and CIFAR-100 datasets, with E1 representing the proposed approach. First, we study the efficacy of the

*Table A1.* **CIFAR-10, CIFAR-100**: Ablation experiments on ResNet-18 architecture to highlight the importance of various aspects in the proposed defense OA-AT. Performance (%) against attacks with different $\varepsilon$ bounds is reported.

| Method | CIFAR-10 | | | | CIFAR-100 | | | |
|---|---|---|---|---|---|---|---|---|
| | Clean | GAMA (8/255) | GAMA (16/255) | Square (16/255) | Clean | GAMA (8/255) | GAMA (16/255) | Square (16/255) |
| **E1**: OAAT (Ours) | 80.24 | **51.40** | 22.73 | 31.16 | 60.27 | **26.41** | 10.47 | 14.60 |
| **E2**: LPIPS weight = 0 | 78.47 | 50.60 | 24.05 | 31.37 | 58.47 | 25.94 | 10.91 | 14.66 |
| **E3**: Alpha = 1 | 79.29 | 50.60 | 23.65 | 31.23 | 58.84 | 26.15 | 10.97 | 14.89 |
| **E4**: Alpha = 1, LPIPS weight = 0 | 77.16 | 50.49 | **24.93** | **32.01** | 57.77 | 25.92 | **11.33** | **15.03** |
| **E5**: Using Current model (without WA) for LPIPS | 80.50 | 50.75 | 22.90 | 30.76 | 59.54 | 26.23 | 10.50 | 14.86 |
| **E6**: Using Discriminator instead of LPIPS (OI Attack) | 80.56 | 50.75 | 22.13 | 31.17 | 58.84 | 26.35 | 10.64 | 14.82 |
| **E7**: Without 2*eps perturbations for AWP | 79.96 | 50.50 | 22.61 | 30.60 | 60.18 | 26.27 | 10.15 | 14.20 |
| **E8**: Increasing epsilon from the beginning | 80.34 | 50.77 | 22.57 | 30.80 | **60.51** | 26.34 | 10.37 | 14.61 |
| **E9**: Maximizing KL div in the AWP step | 81.19 | 49.77 | 21.17 | 29.39 | 59.48 | 25.03 | 7.93 | 13.34 |
| **E10**: Without AutoAugment | 80.24 | **51.40** | 22.73 | 31.16 | 58.08 | 25.81 | 10.40 | 14.31 |
| **E11**: With AutoAugment (p=0.5) | 81.59 | 50.40 | 21.59 | 30.84 | 60.27 | **26.41** | 10.47 | 14.60 |
| **E12**: With AutoAugment (p=1) | **81.74** | 48.15 | 18.92 | 28.31 | 60.19 | 25.32 | 9.24 | 13.78 |

LPIPS metric in generating Oracle-Invariant attacks. In experiment E2, we train a model without LPIPS by setting its coefficient to zero. While the resulting model achieves a slight boost in robust accuracy at $\varepsilon = 16/255$ due to the use of stronger attacks for training, there is a considerable drop in clean accuracy, and a corresponding drop in robust accuracy at $\varepsilon = 8/255$ as well. We observe a similar trend by setting the value of $\alpha$ to 1 as shown in E3, and by combining E2 and E3 as shown in E4. We note that E4 is similar to standard adversarial training, where the model attempts to learn consistent predictions in the $\varepsilon$ ball around every data sample. While this works well for large $\varepsilon$ attacks ($\varepsilon = 16/255$), it leads to poor clean accuracy as shown in the first partition of Table-A2.

As discussed in Sec.4, we maximize loss on $x_i + 2 \cdot \widetilde{\delta}_i$ (where $\widetilde{\delta}_i$ is the attack) in the additional weight perturbation step. We present results by using the standard $\varepsilon$ limit for the weight perturbation step as well, in E7. This leads to a drop across all metrics, indicating the importance of using large magnitude perturbations in the weight perturbation step for producing a flatter loss surface that leads to better generalization to the test set. Different from the standard TRADES formulation, we maximize Cross-Entropy loss for attack generation in the proposed method. From E9 we note that the use of KL divergence leads to a drop in robust accuracy since the KL divergence based attack is weaker. This is consistent with the observation by Gowal et al. [12].

## A4. Detailed Results

In Tables-A2 and A3, we present results of different defense methods such as AWP-TRADES [30], TRADES [31], PGD-AT [18], ExAT [21], ATES [22] and FAT [1], evaluated across a wide range of adversarial attacks. We present evaluations on the Black-Box FGSM attack [11] and a suite of White-Box attacks, on $\ell_\infty$ constraint sets of different radii:

$8/255$, $12/255$ and $16/255$. The white-box evaluations consist of the single-step Randomized-FGSM (R-FGSM) attack [27], the GAMA PGD-100 attack [23] and AutoAttack [8], with the latter two being amongst the strongest of attacks known to date. Lastly, we also present evaluations on the Square attack [2] for $\varepsilon = 12/255$ and $16/255$ in order to evaluate performance on Oracle-Invariant samples at large perturbation bounds.

**CIFAR-10:** To enable a fair comparison of the proposed approach with existing methods, we present comprehensive results of various defenses trained with different attack strengths in Table-A2. In the first partition of the table, we present baselines trained using attacks constrained within an $\ell_\infty$ bound of $16/255$. While these models do achieve competitive robustness on adversaries of attack strength $\varepsilon = 8/255$, $12/255$ and $16/255$, they achieve significantly lower accuracy on clean samples which limits their use in practical scenarios. Thus, for better comparative analysis that accounts for the robustness-accuracy trade-off, we present results of the existing methods with hyperparameters and attack strengths tuned to achieve the best robust performance, while maintaining clean accuracy close to $80\%$ as commonly observed on the CIFAR-10 dataset on ResNet-18 architecture, in the second partition of Table-A2. We observe that the proposed method OA-AT consistently outperforms other approaches on all three metrics described in Sec.3.3, by achieving enhanced performance at $\varepsilon = 8/255$ and $16/255$, while striking a favourable robustness-accuracy trade-off as well. The proposed defense achieves better robust performance even on the standard $\ell_\infty$ constraint set of $8/255$ when compared to existing approaches, despite being trained on larger perturbations sets.

**CIFAR-100:** In Table-A3, we present results on models trained on the highly-challenging CIFAR-100 dataset. Since this dataset contains relatively fewer training images per class, we seek to enhance performance further by incorpo-

*Table A2.* **CIFAR-10**: Performance (%) of the proposed defense OA-AT against attacks with different $\varepsilon$ bounds, when compared to the following baselines: AWP [30], ExAT [21], TRADES [31], ATES [22], PGD-AT [18] and FAT [1]. AWP [30] is the strongest baseline. The first partition shows defenses trained on $\varepsilon = 16/255$. Training on large perturbation bounds results in very poor Clean Accuracy. The second partition consists of baselines tuned to achieve clean accuracy close to 80%. These are sorted by AutoAttack accuracy [8] (AA 8/255). The proposed defense achieves significant gains in accuracy across all attacks.

| Method | Attack $\varepsilon$ (Training) | Clean | FGSM (BB) (8/255) | R-FGSM (8/255) | GAMA (8/255) | AA (8/255) | FGSM (BB) (12/255) | R-FGSM (12/255) | GAMA (12/255) | Square (12/255) | FGSM (BB) (16/255) | R-FGSM (16/255) | GAMA (16/255) | Square (16/255) |
|---|---|---|---|---|---|---|---|---|---|---|---|---|---|---|
| TRADES | 16/255 | 75.30 | 73.26 | 53.10 | 35.64 | 35.12 | 72.13 | 44.27 | 20.24 | 30.11 | 70.76 | 36.99 | 10.10 | 18.87 |
| AWP | 16/255 | 71.63 | 69.71 | 54.53 | 40.85 | 40.55 | 68.65 | 47.13 | 27.06 | 34.42 | 67.42 | 40.89 | 15.92 | 24.16 |
| PGD-AT | 16/255 | 64.93 | 63.65 | 55.47 | 46.66 | 46.21 | 62.81 | 51.05 | 36.95 | 40.53 | 61.70 | 46.40 | 26.73 | 32.25 |
| FAT | 16/255 | 75.27 | 73.44 | 60.25 | 47.68 | 47.34 | 72.22 | 53.17 | 34.31 | 39.79 | 70.73 | 46.88 | 22.93 | 29.47 |
| ExAT+AWP | 16/255 | 75.28 | 73.27 | 60.02 | 47.63 | 47.46 | 71.81 | 52.38 | 34.42 | 39.62 | 70.47 | 45.39 | 22.61 | 28.79 |
| ATES | 16/255 | 66.78 | 65.60 | 56.79 | 47.89 | 47.52 | 64.64 | 51.71 | 37.47 | 42.07 | 63.75 | 47.28 | 26.50 | 32.55 |
| ExAT + PGD | 16/255 | 72.04 | 70.68 | 59.99 | 49.24 | 48.80 | 69.66 | 53.96 | 36.68 | 41.93 | 68.04 | 48.37 | 23.01 | 30.21 |
| FAT | 12/255 | 80.27 | 77.87 | 61.46 | 45.42 | 45.13 | 76.69 | 52.33 | 29.08 | 36.71 | 74.79 | 44.56 | 16.18 | 24.59 |
| FAT | 8/255 | **84.36** | 82.20 | 64.06 | 48.41 | 48.14 | 80.32 | 55.41 | 29.39 | 39.48 | 78.13 | 47.50 | 15.18 | 25.07 |
| ATES | 8/255 | 84.29 | **82.39** | **65.66** | 49.14 | 48.56 | **80.81** | 55.59 | 29.36 | 40.68 | **78.48** | 47.03 | 14.70 | 25.88 |
| PGD-AT | 8/255 | 81.12 | 78.94 | 63.48 | 49.03 | 48.58 | 77.19 | 54.42 | 30.84 | 40.82 | 74.37 | 46.28 | 15.77 | 26.47 |
| PGD-AT | 10/255 | 79.38 | 77.89 | 62.78 | 49.28 | 48.68 | 76.60 | 54.76 | 32.40 | 41.46 | 74.75 | 47.46 | 18.18 | 28.29 |
| AWP | 10/255 | 80.32 | 77.87 | 62.33 | 49.06 | 48.89 | 76.33 | 53.83 | 32.88 | 40.27 | 74.13 | 45.51 | 19.17 | 27.56 |
| ATES | 10/255 | 80.95 | 79.22 | 63.95 | 49.57 | 49.12 | 77.77 | 55.37 | 32.44 | 42.21 | 75.51 | 48.12 | 18.36 | 29.07 |
| TRADES | 8/255 | 80.53 | 78.58 | 63.69 | 49.63 | 49.42 | 77.20 | 55.48 | 33.32 | 40.94 | 75.05 | 47.92 | 19.27 | 27.82 |
| ExAT + PGD | 11/255 | 80.68 | 79.07 | 63.58 | 50.06 | 49.52 | 77.98 | 55.92 | 32.47 | 41.10 | 76.12 | 48.37 | 17.81 | 27.23 |
| ExAT + AWP | 10/255 | 80.18 | 78.04 | 63.15 | 49.87 | 49.69 | 76.34 | 54.64 | 33.51 | 41.04 | 74.37 | 46.54 | 20.04 | 28.40 |
| AWP | 8/255 | 80.47 | 78.22 | 63.32 | 50.06 | 49.87 | 76.88 | 54.61 | 33.47 | 41.05 | 74.42 | 46.16 | 19.66 | 28.51 |
| **OA-AT (Ours)** | 16/255 | 80.24 | 78.54 | 65.00 | **51.40** | **50.88** | 77.34 | **57.68** | **36.01** | **43.20** | 75.72 | **51.13** | **22.73** | **31.16** |
| Gain w.r.t. AWP | | −0.23 | +0.32 | +1.68 | +1.34 | +1.01 | +0.46 | +3.07 | +2.54 | +2.15 | +1.30 | +4.97 | +3.07 | +2.65 |

*Table A3.* **CIFAR-100**: Performance (%) of the proposed defense OA-AT against attacks with different $\varepsilon$ bounds, when compared to the following baselines: AWP [30], ExAT [21], TRADES [31], ATES [22], PGD-AT [18] and FAT [1]. AWP [30] is the strongest baseline. The baselines are sorted by AutoAttack accuracy [8] (AA 8/255). The proposed defense achieves significant gains in accuracy against the strongest attacks across all $\varepsilon$ bounds. Since the proposed defense uses AutoAugment [9] as the augmentation strategy, we present results on the strongest baseline AWP [30] with AutoAugment as well.

| Method | Attack $\varepsilon$ (Training) | Clean | FGSM (BB) (8/255) | R-FGSM (8/255) | GAMA (8/255) | AA (8/255) | FGSM (BB) (12/255) | R-FGSM (12/255) | GAMA (12/255) | Square (12/255) | FGSM (BB) (16/255) | R-FGSM (16/255) | GAMA (16/255) | Square (16/255) |
|---|---|---|---|---|---|---|---|---|---|---|---|---|---|---|
| FAT | 8/255 | 56.61 | 52.10 | 34.76 | 23.36 | 23.20 | 49.54 | 27.77 | 13.96 | 18.21 | 46.01 | 22.52 | 8.30 | 11.56 |
| TRADES | 8/255 | 58.27 | 54.33 | 36.20 | 23.67 | 23.47 | 51.64 | 28.55 | 13.88 | 18.46 | 48.46 | 22.78 | 8.31 | 11.89 |
| PGD-AT | 8/255 | 57.43 | 53.71 | 37.66 | 24.81 | 24.33 | 50.90 | 30.07 | 13.51 | 19.62 | 47.43 | 23.18 | 7.40 | 11.64 |
| ATES | 8/255 | 57.54 | 53.62 | 37.05 | 25.08 | 24.72 | 50.84 | 29.18 | 13.75 | 19.42 | 47.35 | 22.89 | 7.59 | 11.40 |
| ExAT-PGD | 9/255 | 57.46 | 53.56 | 38.48 | 25.25 | 24.93 | 51.43 | 30.60 | 15.12 | 20.40 | 48.15 | 24.21 | 8.37 | 12.47 |
| ExAT-AWP | 10/255 | 57.76 | 53.46 | 37.84 | 25.55 | 25.27 | 50.42 | 30.39 | 14.98 | 19.72 | 46.99 | 24.48 | 9.07 | 12.68 |
| AWP | 8/255 | 58.81 | 54.13 | 37.92 | 25.51 | 25.30 | 50.72 | 30.40 | 14.71 | 19.82 | 46.66 | 23.96 | 8.68 | 12.44 |
| AWP (with AutoAug.) | 8/255 | 59.88 | 55.62 | 39.10 | 25.81 | 25.52 | 52.75 | 31.11 | 14.80 | 20.24 | 49.44 | 24.99 | 8.72 | 12.80 |
| **OA-AT (Ours) (with AutoAug.)** | 16/255 | **60.27** | **56.27** | **40.24** | **26.41** | **26.00** | **53.86** | **33.78** | **16.28** | **21.47** | **51.11** | **28.02** | **10.47** | **14.60** |
| Gain w.r.t. AWP (with AutoAug.) | | +0.39 | +0.65 | +1.14 | +0.60 | +0.48 | +1.11 | +2.67 | +1.48 | +1.23 | +1.67 | +3.03 | +1.75 | +1.80 |

rating the augmentation technique, AutoAugment [9, 24]. To enable fair comparison, we incorporate AutoAugment for the strongest baseline, AWP [30] as well. We observe that the proposed method consistently performs better than existing approaches by significant margins, both in terms of clean accuracy, as well as robustness against adversarial attacks conforming to the three distinct constraint sets. Further, this also confirms that the proposed method scales well to large, complex datasets, while maintaining a consistent advantage in performance compared to other approaches.

## A5. Gradient Masking Checks

As discussed by Athalye et al. [3], we present various checks to ensure the absence of Gradient Masking in the proposed defense. In Fig.A4(a,c), we observe that the accuracy of the proposed defense on the CIFAR-10 and CIFAR-100 datasets monotonically decreases to zero against 7-step PGD white-box attacks as the perturbation budget is increased. This shows that gradient based attacks indeed serve as a good indicator of robust performance, as strong adversaries of large perturbation sizes achieve zero accuracy, indicating the absence of gradient masking. In Fig.A4(b,d), we plot the Cross-Entropy loss against FGSM attacks with varying perturbation budget. We observe that the loss increases linearly, thereby suggesting that the first-order Taylor approximation to the loss surface indeed remains effective in the local neighbourhood of sample images, again indicating the absence of gradient masking.

We verify that the model achieves higher robust accuracy against weaker Black-box attacks, as compared to strong gradient based attacks such as GAMA or AutoAttack in

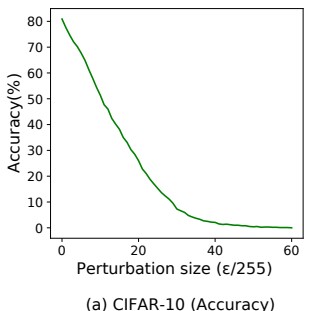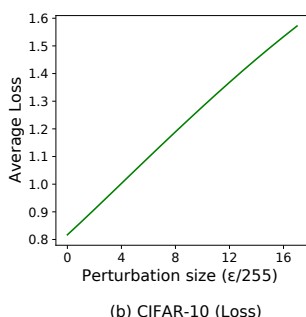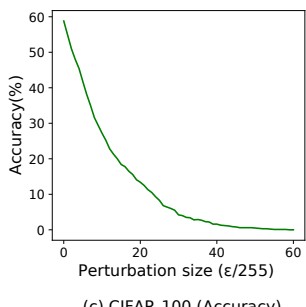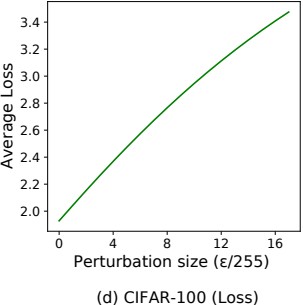

|                        |                     |                    |                       |
|------------------------|---------------------|--------------------|-----------------------|
| (a) CIFAR-10 (Accuracy) | (b) CIFAR-10 (Loss) | (c) CIFAR-100 (Accuracy) | (d) CIFAR-100 (Loss) |

*Figure A4.* Accuracy and Loss plots on a 1000-sample class-balanced subset of the respective test-sets of CIFAR-10 and CIFAR-100 datasets. (a, c) Plots showing the trend of Accuracy (%) against PGD-7 step attacks across variation in attack perturbation bound ($\varepsilon$) on CIFAR-10 and CIFAR-100 datasets with ResNet-18 architecture. As the perturbation bound increases, accuracy against white-box attacks goes to 0, indicating the absence of gradient masking [3] (b, d) Plots showing the variation of Cross-Entropy Loss on FGSM attack [11] against variation in the attack perturbation bound ($\varepsilon$) on CIFAR-10 and CIFAR-100 datasets. As the perturbation bound increases, loss increases linearly, indicating the absence of gradient masking [3]

*Table A4.* **Evaluation against various attacks with a perturbation bound of** $\varepsilon = 8/255$ **on CIFAR-10**: Performance (%) of the proposed defense OA-AT against various attacks (sorted by Robust Accuracy) to ensure the absence of gradient masking. [†]Includes 5000-queries of Square attack.

| Attack | No. of Steps | No. of restarts | Robust Accuracy (%) |
|--------|:---:|:---:|:---:|
| AutoAttack[†] [8] | 100 | 20 | **50.88** |
| GAMA-MT [23] | 100 | 5 | 50.90 |
| ODS (98 +2 steps) [26] | 100 | 100 | 50.94 |
| MDMT attack [15] | 100 | 10 | 51.19 |
| Logit-Scaling attack [4, 13] | 100 | 20 | 51.26 |
| GAMA-PGD [23] | 100 | 1 | 51.40 |
| MD attack [15] | 100 | 1 | 51.47 |
| PGD-50 (1000 RR) [18] | 50 | 1000 | 55.37 |
| PGD-1000 [18] | 1000 | 1 | 56.15 |

Tables-A2,A3. We also observe that adversaries that conform to larger constraint sets are stronger than their counterparts that are restricted to smaller epsilon bounds, as expected.

In Table-A4, we perform exhaustive evaluations using various attack techniques to further verify the absence of gradient masking. In addition to AutoAttack [8] which in itself consists of an ensemble of four attacks- AutoPGD with Cross-Entropy and Difference-of-Logits loss, the FAB attack [7] and Square Attack [2], we present evaluations against strong multi-targeted attacks such as GAMA-MT [23] and the MDMT attack [15] which specifically target other classes during optimization. We also consider the untargeted versions of the latter two attacks, the GAMA-PGD and MD attack respectively. We also present robustness against the ODS attack [26] with 100 restarts, which diversifies the input random noise based on the output predictions in order to obtain results which are less dependent on the sampled random noise used for attack initialization. Next, the Logit-Scaling attack [4, 13] helps yield robust evaluations that are less dependent on the exact scale of output logits predicted by the network, and is seen to be effective

on some defenses which exhibit gradient masking. However, we observe that the proposed method is robust against all such attacks, with the lowest accuracy being attained on the AutoAttack ensemble.

Furthermore, we evaluate the model on PGD 50-step attack run with 1000 restarts. The robust accuracy saturates with increasing restarts, with the final accuracy still being higher than that achieved on AutoAttack. Lastly, we observe that the PGD-1000 attack is not very strong, confirming that the accuracy does not continually decrease as the number of steps used in the attack increases. Thus, we observe that the proposed approach is robust against a diverse set of attack methods, thereby confirming the absence of gradient masking and verifying that the model is truly robust.