# OpenReview forum: "Towards Achieving Adversarial Robustness Beyond Perceptual Limits"
_ICML.cc/2021/Workshop/AML — ICML 2021 Workshop AML Poster_

### Official Review · Reviewer_eti6 · 2021-06-19
**The paper clearly defines the Oracle-Invariant set and the Oracle-Sensitive set. The proposed OA-AT method is an effective defense training algorithm.**

**Rating:** Accept
**Confidence:** 4

**Review:**

Pros:
1. The writing is good.
2. The research domain is clearly stated. The difference between Oracle-Invariant set and the Oracle-Sensitive set is clearly stated.
3. Experiments show that the proposed OA-AT is an effective algorithm to train defense model.
4. The experimental results are sufficient.
Cons:
1. Reference format is not standard.

---

### Decision · Program_Chairs · 2021-06-21

**Decision:**

Accept (Poster)

**Comment:**

A good paper in terms of writing, experiments, and techniques, as noted by the reviewer.